# Abietane Diterpenoids from the Bark of *Cryptomeria japonica* and Their Antifungal Activities against Wood Decay Fungi

**DOI:** 10.3390/plants13091197

**Published:** 2024-04-25

**Authors:** Chi-I Chang, Cheng-Chi Chen, Sheng-Yang Wang, Yueh-Hsiung Kuo

**Affiliations:** 1Department of Biological Science and Technology, National Pingtung University of Science and Technology, Pingtung 912, Taiwan; changchii@mail.npust.edu.tw; 2Traditional Herbal Medicine Research Center, Taipei Medical University Hospital, Taipei 110, Taiwan; 3Department of Chemistry, National Taiwan University, Taipei 106, Taiwan; changchii@yahoo.com.tw; 4Department of Forestry, National Chung-Hsing University, Taichung 402, Taiwan; taiwanfir@dragon.nchu.edu.tw; 5Agricultural Biotechnology Research Center, Academia Sinica, Taipei 115, Taiwan; 6Department of Chinese Pharmaceutical Sciences and Chinese Medicine Resources, College of Pharmacy, China Medical University, Taichung 404, Taiwan; 7Department of Biotechnology, Asia University, Taichung 413, Taiwan; 8Chinese Medicine Research Center, China Medical University, Taichung 404, Taiwan

**Keywords:** Cupressaceae, *Cryptomeria japonica*, bark, abietane, antifungal

## Abstract

Phytochemical investigation of the bark of *Cryptomeria japonica* led to the isolation of five new abietane diterpenoids, 5-*epi*-12-hydroxy-6-*nor*-5,6-secoabieta-8,11,13-trien-7,5-olide (**1**), 12-hydroxy-6β-methoxy-6,7-secoabieta-8,11,13-trien-7,6-olide (**2**), 6β,12-dihydroxy-7,8-secoabieta-8,11,13-trien-7,8-olide (**4**), 5,12-dihydroxy-7,8-secoabieta-8,11,13-trien-7,8-olide (**5**), and 5α,8-epoxy-12-hydroxy-7,8-secoabieta-8,11,13-trien-7-al (**6**), together with one known abietane diterpenoid, obtuanhydride (**3**). Their structures were elucidated by analysis of spectroscopic data and comparison with the spectral data of known analogs. At the concentration of 100 μg/mL, compounds **4**, **5,** and **6** inhibited antifungal activities against wood decay fungi activity by 18.7, 37.2, and 46.7%, respectively.

## 1. Introduction

*Cryptomeria japonica* D. Don is a massive evergreen coniferous tree belonging to the monospecific genus *Cryptomeria* in the cypress family Cupressaceae. It is endemic to Japan, known as Japanese cedar or sugi in Japanese [1], and is the main forestry tree species in Japan. This conifer is widely distributed in warm and cool temperate climates. The wood of *C. japonica* is used as a raw material to produce building materials and wood products. Due to its excellent properties, such as aromatic, reddish-pink in color, soft, lightweight yet sturdy, waterproof, and resistant to natural decay, it has become one of the most commercially important plantation forest tree species in several Asian countries, including Japan, Taiwan, Korea, China, India, and Nepal. Researchers have identified a variety of terpenoids, including monoterpenoids, sesquiterpenoids, and diterpenoids from the leaves, heartwood, and barks of this plant [2,3,4,5,6,7,8,9,10,11,12,13,14,15,16,17,18,19,20,21,22,23,24]. Additionally, the crude extracts and secondary metabolites of this plant have been proven to possess a wide range of bioactivities, including cytotoxic [23], antifungal [24], antibacterial [23], antioxidant [25], anti-inflammatory [26], and insect antifeedant [27] and repellent [18] properties. During our ongoing search for new and bioactive metabolites from the bark of *C. japonica*, three sesquarterpenoids [28,29] and ten abietane-type diterpenoids had already been reported by us [29].

*Phellinus noxius* is an aggressive and destructive pathogen. Trees infected by *P. noxius* could develop brown root disease. The mycelium of *P. noxius* mainly grows in tree roots or stump tissue and cannot grow freely in soil. Infection attacks the roots of plants when healthy tree roots come into contact with the roots of infected trees or soil with stump tissue. After the tree is infected, the propagation and spread of the brown root pathogen in the tree will cause the decay of the root wood tissue, making the root fragile, affecting the root system’s ability to absorb nutrients from the soil, and even making it prone to lodging due to reduced support. The most significant impact is the decay and death of trees, resulting in fewer trees in urban areas and reduced horticultural and forestry productivity. This disease is difficult to control because the fungus can survive in the soil for many years. The presence of this disease usually indicates extensive wood decay, which can cause structural damage and subsequent tree failure [30].

The purpose of this study was to evaluate whether the diterpenoids of *C. japonica* bark possessed the anti-brown Rhizobium activity. Herein, we described the isolation, structural elucidation, and antifungal activities against *P. noxius* of compounds **1**–**6** (Figure 1).

## 2. Results and Discussion

The EtOAc soluble portion partitioned from methanol extracts of the bark of *C. japonica* was subjected to repeated chromatography on silica gel followed by semipreparative NP-HPLC. Five new abietane diterpenoids, 5-*epi*-12-hydroxy-6-*nor*-5,6-secoabieta-8,11,13-trien-7,5-olide (**1**), 12-hydroxy-6β-methoxy-6,7-secoabieta-8,11,13-trien-7,6-olide (**2**), 6β,12-dihydroxy-7,8-secoabieta-8,11,13-trien-7,8-olide (**4**), 5,12-dihydroxy-7,8-secoabieta-8,11,13-trien-7,8-olide (**5**), and 5α,8-epoxy-12-hydroxy-7,8-secoabieta-8,11,13-trien-7-al (**6**), together with one known abietane diterpenoid, obtuanhydride (**3**) [31], were obtained. 

The HR-EI-MS of **1** gave a molecular ion at *m*/*z* 302.1880, establishing the molecular formula of **1** as C_19_H_26_O_3_, with seven degrees of unsaturation. The UV maximum (271 nm) and IR absorptions (1679, 1606, and 1513 cm^−1^) of **1** indicated the presence of the benzoyl moiety [32] (see Appendix A). An IR absorption for a hydroxyl group (3297 cm^−1^) was also observed. The resonances in the ^1^H NMR spectrum of **1** (Table 1) for three tertiary-linked methyls [δ_H_ 0.35, 1.08, and 1.28 (each 3H, s, Me-18, Me-19, and Me-20)], two *para*-oriented aromatic protons [δ_H_ 6.65 (1H, s) and 7.91 (1H, s)], an isopropyl group [δ_H_ 1.25 (3H, d, *J* = 7.2 Hz), 1.26 (3H, d, *J* = 7.2 Hz), and 3.14 (1H, sept, *J* = 7.2 Hz)], and a phenolic proton [δ_H_ 5.71 (1H, s, D_2_O exchange)] suggested that **1** was a dehydroabietane-like diterpene [32] (Kuo and Yu, 1996). The ^13^C NMR and DEPT spectra of **1** indicated the presence of 19 carbons, consisting of five methyl, three aliphatic methylene, two aliphatic methine, two aliphatic quaternary, two olefinic methine, four quaternary olefinic, and one conjugated lactone carbonyl carbons (Table 2). Accounting for the seven degrees of unsaturation attributing from the rings A and C and a carbonyl group, the remaining one degree of unsaturation hinted that the conjugated lactone carbonyl (δ_C_ 164.6) was located at C-7 and linked via an oxygen atom to C-5 (δ_C_ 91.6). The HMBC correlations between H-5 (δ_H_ 4.05) and C-4, C-7, C-9, C-10, C-18, and C-20 confirmed the above proposal. Furthermore, H-5 showed the NOESY correlation with Me-20 (δ_H_ 1.28), confirming the *cis*-ring junction between rings A and B. Both the phenyl group (ring C) and Me-18 were situated in axial orientation, resulting in an unusual upshifted Me-18 proton signal (δ_H_ 0.35) due to receiving an anisotropic effect from the phenyl group. Additionally, the HMBC correlations between H-15/C-12 and C-13; Me-16/C-13; Me-18/C-3; Me-19/C-5; and Me-20/C-1, C-5, and C-9 helped to construct the planar structure of **1**. The relative configurations of stereogenic C-atoms in the tricyclic rings were determined by significant NOE correlations between H_α_-1 (δ_H_ 2.36)/H-11, H-5/Me-19, and H-5/Me-20 in the NOESY spectrum (Figure 2). Thus, the structure of **1** was determined as 5-*epi*-12-hydroxy-6-*nor*-5,6-secoabieta-8,11,13-trien-7,5-olide. Complete ^1^H and ^13^C NMR chemical shifts were established by ^1^H-^1^H COSY, HMQC, HMBC, and NOESY spectra.

The UV maximum (271 nm) and IR absorptions (1679, 1606, and 1460 cm^−1^) of **2** indicated the presence of the benzoyl moiety [32]. An IR absorption at 3337 cm^−1^ for the hydroxyl group was also observed. The molecular formula was established to be C_21_H_30_O_4_ from its HR-EI-MS molecular ion at *m*/*z* 346.2150 and its ^13^C NMR data, indicating seven degrees of unsaturation. The ^1^H NMR spectrum of **2** (Table 1) showed resonances for three tertiary-linked methyls [δ_H_ 0.91, 1.12, and 1.44 (each 3H, s, Me-18, Me-19, and Me-20)], two *para*-oriented aromatic protons [δ_H_ 6.69 (1H, s) and 7.65 (1H, s)], an isopropyl group [δ_H_ 1.25 (3H × 2, d, *J* = 6.8 Hz) and 3.11 (1H, sept, *J* = 6.8 Hz)], an oxymethine [δ_H_ 5.11 (1H, d, 1.6 Hz)], a methoxy [δ_H_ 3.34 (3H, s)], and a phenolic proton [δ_H_ 5.24 (1H, s, D_2_O exchange)]. A total of 21 carbon signals were observed in the ^13^C NMR spectrum of **2** and were differentiated by DEPT experiments as five aliphatic methyl, three aliphatic methylene, two aliphatic methine, two aliphatic quaternary, one oxygenated methine, two olefinic methine, four quaternary olefinic, one methoxy, and one lactone carbonyl carbons. From the above evidence, compound **2** was suggested as a dehydroabietane diterpene [32]. After subtracting the 6 degrees of unsaturation derived from the rings A and C and the carbonyl group, the remaining one degree of unsaturation, together with the downshifted H-14 [δ_H_ 7.65 (1H, s)], suggested that the conjugated lactone carbonyl (δ_C_ 170.1) was located at C-7 and linked via an oxygen atom to the hemiacetal carbon, C-6 (δ_C_ 105.1). The hemiacetal proton, H-6 [δ_H_ 5.11 (1H, d, *J* = 1.6 Hz, H-6)], showed both ^1^H-^1^H COSY correlation with H-5 with a small coupling constant, 1.6 Hz, and NOESY correlations with Me-18 and Me-19 confirmed the methoxyl group attached to C-6 in β orientation (Figure 2). In addition, H-5 showed a NOESY correlation with Me-18 (δ_H_ 0.91), while no NOESY correlation with Me-19 implied the *trans*-ring junction between rings A and B. From the above evidence, compound **2** was thus formulated as 12-hydroxy-6β-methoxy-6,7-secoabieta-8,11,13-trien-7,6-olide.

The molecular formula of **4** was assigned as C_20_H_28_O_4_ by HR-EI-MS at *m*/*z* 332.1978, representing seven degrees of unsaturation. The IR absorptions indicated the presence of a hydroxyl (3429 cm^−1^) group and a lactone carbonyl group (1725 cm^−1^). The ^1^H NMR spectrum of **4** (Table 1) displayed the signals for three tertiary-linked methyls [δ_H_ 1.00, 1.06, and 1.45 (each 3H, s, Me-18, Me-19, and Me-20)], one oxymethine [δ_H_ 4.53 (1H, s, H-6)], two *para*-oriented aromatic protons [δ_H_ 6.71 (1H, s) and 6.97 (1H, s)], an isopropyl group on the benzene ring [δ_H_ 1.22 (3H, d, *J* = 6.8 Hz), 1.23 (3H, d, *J* = 6.8 Hz), and 3.11 (1H, sept, *J* = 6.8 Hz)], and a phenolic proton [δ_H_ 4.94 (1H, s, D_2_O exchange)]. A total of 20 carbon signals were found in the ^13^C NMR spectrum of **4** and were assigned by a DEPT experiment as five aliphatic methyl, three aliphatic methylene, two aliphatic methine, two aliphatic quaternary, one oxygenated methine, two olefinic methine, four quaternary olefinic, and one lactone carbonyl carbons. From the above evidence, compound **4** was proposed to be a dehydroabietane diterpene [32]. A downshifted oxymethine [δ_H_ 4.53 (1H, s, H-6)] neighboring to the carbonyl group (δ_C_ 171.1) and an upshifted phenyl proton H-14 [δ_H_ 6.97 (1H, s)] were observed, which suggested that the carbonyl group was situated at C-7, linking to C-8 via an oxygen atom. The *trans*-ring junction between rings A and B was confirmed by the NOESY correlation between H-5/H-6, H-6/Me-18, H-6/Me-19, Me-19/Me-20, and H-11/Me-20 (Figure 2). Furthermore, the HMBC correlations between H-6/C-4 and C-10 and the NOESY correlations between H-6/Me-18 and Me-19 hinted at the hydroxyl group attached to C-6 in β-axial orientation. Thus, compound **4** was identified as 6β,12-dihydroxy-7,8-secoabieta-8,11,13-trien-7,8-olide. 

The HR-EI-MS of **5** gave a molecular ion at *m*/*z* 332.1977, consistent with the molecular formula of C_20_H_28_O_4_, implying seven degrees of unsaturation. The IR absorptions indicated the presence of a hydroxyl (3416 cm^−1^) group and a lactone carbonyl group (1699 cm^−1^). The ^1^H NMR spectrum of **5** (Table 1) displayed the signals for three tertiary-linked methyls [δ_H_ 1.16, 1.20, and 1.37 (each 3H, s, Me-19, Me-18, and Me-20)], two *para*-oriented aromatic protons [δ_H_ 6.46 (1H, s) and 6.72 (1H, s)], an isopropyl group on the benzene ring [δ_H_ 1.20 (3H, d, *J* = 6.8 Hz), 1.21 (3H, d, *J* = 6.8 Hz), and 3.13 (1H, sept, *J* = 6.8 Hz)], and a typical AB-type methylene neighboring to a carbonyl group [δ_H_ 2.57 (1H, d, *J* = 16.8 Hz) and 2.61 (1H, d, *J* = 16.8 Hz)]. A total of 20 carbon signals were found in the ^13^C NMR spectrum of **5** and were assigned by a DEPT experiment as five aliphatic methyl, four aliphatic methylene, one aliphatic methine, two aliphatic quaternary, one oxygenated quaternary, two olefinic methine, four quaternary olefinic, and one lactone carbonyl carbons. Compound **5** showed identical NMR characteristics to that of **4**, and the only difference was in the ring B part. The hydroxyl group was attached to C-5 in **5** instead of C-6 in **4**, which was assured by the HMBC correlations between H-6 with C-4, C-5, C-7, and C-10. The NOESY correlation between H_α_-6 (δ_H_ 2.57)/Me-18, H_β_-6 (δ_H_ 2.61)/Me-19, Me-20/Me-19, and Me-20/H-11 assured the *trans*-ring junction between rings A and B (Figure 2). Thus, the structure of **5** was determined as 5,12-dihydroxy-7,8-secoabieta-8,11,13-trien-7,8-olide.

The HR-EI-MS of compound **6** showed an [M]^+^ ion at *m*/*z* 316.2033, which was consistent with the molecular formula C_20_H_28_O_3_, indicating seven degrees of unsaturation. The IR spectrum indicated the presence of a hydroxyl (3423 cm^−1^) group and an aldehyde carbonyl group (1706 cm^−1^). In the ^1^H NMR spectra of **6** (Table 1), the signals for three tertiary-linked methyls [δ_H_ 1.05, 1.17, and 1.38 (each 3H, s, Me-19, Me-18, and Me-20], two *para*-oriented aromatic protons [δ_H_ 6.38 (1H, s) and 6.65 (1H, s)], an isopropyl group on the benzene ring [δ_H_ 1.17 (3H, d, *J* = 6.8 Hz), 1.18 (3H, d, *J* = 6.8 Hz), and 3.13 (1H, sept, *J* = 6.8 Hz)], a phenolic proton [δ_H_ 5.11 (1H, s, D_2_O exchange)], and an A_2_X coupling system of a methylene neighboring to an aldehyde group [δ_H_ 2.61 (2H, d, *J* = 3.6 Hz) and 9.38 (1H, t, *J* = 3.6 Hz)]. The ^13^C-NMR spectrum of **6** revealed twenty skeletal carbon resonances, including five aliphatic methyl, four aliphatic methylene, one aliphatic methine, three aliphatic quaternary, two olefinic methine, four quaternary olefinic, and one aldehyde carbonyl carbons. From the above evidence, compound **6** was also proposed to be a dehydroabietane diterpene. An unusual oxygenated quaternary carbon signal (δ_C_ 96.3), together with an upshifted H-14 (δ_H_ 6.65), suggested that C-5 linked to C-8 via an oxygen atom. The HMBC correlations between H-6 (δ_H_ 2.61)/C-7 and C-10 and the NOESY correlations between H-6/Me-18, H-6/Me-19, H-6/Me-20, Me-20/Me-19, and Me-20/H-11 indicated that the 2-oxoethyl moiety was attached to C-5 in β orientation (Figure 2). Therefore, compound **6** was characterized as 5α,8-epoxy-12-hydroxy-7,8-secoabieta-8,11,13-trien-7-al.

Compound **3** was identified as a known abietane diterpenoid, obtuanhydride (**3**), by comparing their spectral data of NMR and mass with those described in the literature [31].

The brown root rot fungus *P. noxius* was used to evaluate the antifungal activity of *C. japonica*’s compounds. The antifungal indices of the diterpenoids at the dosage of 100 µg/mL are listed in Table 3. Among these compounds, 5,12-dihydroxy-7,8-secoabieta-8,11,13-trien-7,8-olide (**5**) and 5α,8-epoxy-12-hydroxy-7,8-secoabieta-8,11,13-trien-7-al (**6**) exhibited the stronger antifungal activity against *P. noxius* with antifungal indices of 37.2 and 46.7%, respectively, compared to compounds **1**–**4**. The commercial fungicide, didecyldimethylammonium chloride (DDAC), was used as a positive control at a concentration of 10 µg/mL with an antifungal index of 51.1%. 

## 3. Materials and Methods

### 3.1. General Experimental Procedures

Optical rotations were recorded on a Jasco-DIP-180 polarimeter (JASCO Co., Tokyo, Japan). UV and IR spectra were recorded on a Shimadzu UV-1601 (Shimadzu, Kyoto, Japan) and a Perkin-Elmer 983 G (PerkinElmer Ltd., Bucks, UK) spectrophotometer, respectively. ^1^H and ^13^C NMR and 2D NMR spectra were obtained on a Varian-Unity-Plus-400 spectrometer (Varian Inc., Palo Alto, CA, USA). Chemical shifts are referenced to residual solvent signals. EI-MS and HR-EI-MS were obtained on a Jeol-JMS-HX300 mass spectrometer (JEOL Ltd., Tokyo, Japan). Column chromatography (CC) was performed by using Merck Silica gel 60 (230–400 mesh) (Merck, Darmstadt, Germany). Thin-layer chromatography (TLC) analyses were carried out on pre-coated silica gel plates (silica gel 60 F_254_) (Merck, Darmstadt, Germany). HPLC was performed by using a normal phase column (Purospher STAR Si, 5 μm, 250 × 10 mm) (Merck, Darmstadt, Germany) on an LDC Analytical-III system (LDC Analytical, Gelnhausen, Germany). 

### 3.2. Plant Material

The bark of *C. japonica* D. Don was collected in Sitou, Taiwan, in June 2000. A voucher specimen (TCF13443) has been deposited at the Herbarium of the Department of Forestry, NCHU, Taiwan. Species identification was confirmed by Dr. Yen-Hsueh Tseng, Department of Forestry, National Chung-Hsing University (NCHU).

### 3.3. Extraction, Isolation, and Identification

The air-dried bark of *C. japonica* (16.0 kg) was extracted by soaking in MeOH (100 L × 3) at room temperature for 7 days each time in a closed container. The extracts were combined and concentrated under reduced pressure at 45 °C to produce 480 g of a brown crude residue, which was suspended in H_2_O (1 L) and then partitioned sequentially with EtOAc (1 L) and *n*-BuOH (1 L) to afford EtOAc, *n*-BuOH, and H_2_O soluble fractions, respectively. The EtOAc fraction (430 g) was subjected to column chromatography on a silica gel (4.0 kg) column, eluted with a gradient of *n*-hexane–EtOAc, followed by an EtOAc–MeOH gradient of increasing polarity to obtain 11 fractions, fr. 1 (2.6 g), 2 (29.4 g), 3 (47.8 g), 4 (92.4 g), 5 (21.6 g), 6 (18.1 g), 7 (22.5 g), 8 (35.8 g), 9 (19.2 g), 10 (44.2 g), and 11 (72.2 g). Fr. 3 from hexane/AcOEt (9:1) elution (47.8 g) was chromatographed over a silica gel column (7 × 60 cm) using hexane/CH_2_Cl_2_ (1:0–0:1) mixtures to afford nine fractions, 3A–3H. Further purification of subfraction 3F by HPLC afforded **5** (1.1 mg) using hexane/AcOEt (9:1). Fr. 4 from *n*-hexane–EtOAc (4:1) elution was rechromatographed over a silica gel column using a gradient mixture of CH_2_Cl_2_–EtOAc (100:1 to 0:1) to obtain sixteen fractions, 4A–4P. Further purification of subfraction 4I by HPLC afforded **1** (2.8 mg) using *n*-hexane–EtOAc (7:3). Further purification of subfraction 4K by HPLC afforded **6** (3.9 mg) using *n*-hexane–EtOAc (7:3). Further purification of subfraction 4L by HPLC afforded **3** (4.6 mg) using *n*-hexane–EtOAc (7:3). Further purification of subfraction 4M by HPLC afforded **2** (1.4 mg) and **4** (3.1 mg) using *n*-hexane–EtOAc (3:1).

5-*Epi*-12-hydroxy-6-*nor*-5,6-secoabieta-8,11,13-trien-7,5-olide (**1**). Gum; [α]^25^_D_: +20.8 (*c* 0.5, CHCl_3_); IR ν_max_: 3297, 1679, 1606, 1513, 1467, 1367, 1261, 1182, 1122, 1056, 678 cm^−1^; UV (MeOH) λ_max_ (log *ε*): 224 (4.20), 271 (4.05), 295 (3.72) nm; EI-MS *m*/*z* (%): 302 (100) [M]^+^, 287 ([M–CH_3_]^+^, 29), 271 (25), 271 (39), 220 (41), 203 (76), 247 (20), 55 (21); HR-EI-MS [M]^+^
*m*/*z* 302.1880 (calcd for C_19_H_26_O_3_ 302.1883).

12-Hydroxy-6β-methoxy-6,7-secoabieta-8,11,13-trien-7,6-olide (**2**). Gum; [α]^25^_D_: +21.8 (*c* 0.6, CHCl_3_); IR ν_max_: 3337, 1679, 1606, 1580, 1460, 1407, 1261, 1142, 1062, 976 cm^−1^; UV (MeOH) λ_max_ (log *ε*): 221 (4.71), 271 (4.46) nm; EI-MS *m*/*z* (%): 346 ([M]^+^, 12), 314 ([M–HOCH_3_]^+^, 5), 302 (10), 286 (29), 271 (100), 255 (60), 217 (31), 204 (74), 170 (66), 145 (20), 115 (21); HR-EI-MS [M]^+^
*m*/*z* 346.2150 (calcd for C_21_H_30_O_4_ 346.2145).

Obtuanhydride (**3**). Solid; [α]^25^_D_: −17.4 (c 0.18, CHCl_3_); IR ν_max_: 1788, 1734, 1600, 1511, 1248, 1040 cm^−1^; UV (MeOH) λ_max_ (log *ε*): 228 (3.81), 280 (3.52) nm; EI-MS *m*/*z* (%): 344 ([M]^+^, 65), 329 (22), 301 (78), 285 (32), 243 (18), 218 (100), 69 (32); ^1^H-NMR (CDCl_3_): δ 6.81 (s, H-11), 7.62 (s, H-14), 5.99 (br s, Ar-OH), 1.98 (br d, 13.0Hz, H-1), 3.13 (sep, 6.8Hz, H-15), 2.72 (s, H-5), 1.50 (s, H-20), 1.21 (d, 6.8, H-17), 1.23 (d, 6.8, H-16), 1.39 (s, H-19), 1.06 (s,H-18); ^13^C-NMR (CDCl_3_): 42.1 (C-1), 18.8 (C-2), 40.6 (C-3), 33.4 (C-4), 59.2 (C-5), 166.9 (C-6), 165.8 (C-7), 119.5 (C-8), 151.1 (C-9), 40.8 (C-10), 114.3 (C-11), 157.5 (C-12), 133.7 (C-13), 131.8 (C-14), 26.7 (C-15), 22.0 (C-16), 22.2 (C-17), 32.9 (C-18), 22.3 (C-19), 23.0 (C-20).

6β,12-Dihydroxy-7,8-secoabieta-8,11,13-trien-7,8-olide (**4**). Gum; [α]^25^_D_: +30.5 (*c* 1.1, CHCl_3_); IR ν_max_: 3429, 1725, 1507, 1460, 1407, 1241, 1175, 1049, 883, 738 cm^−1^; UV (MeOH) λ_max_ (log *ε*): 284 (3.53) nm; EI-MS *m*/*z* (%): 332 ([M]^+^, 33), 304 ([M–CO]^+^, 100), 289 (10), 273 (17), 271 (20), 299 (17), 203 (13), 192 (8), 179 (17), 152 (11); HR-EI-MS [M]^+^
*m*/*z* 332.1978 (calcd for C_20_H_28_O_4_ 332.1988).

5,12-Dihydroxy-7,8-secoabieta-8,11,13-trien-7,8-olide (**5**). Gum; [α]^25^_D_: +11.5 (*c* 0.3, CHCl_3_); IR ν_max_: 3416, 1699, 1427, 1381, 1261, 1175, 1043, 870,744 cm^−1^_;_ UV (MeOH) λ_max_ (log *ε*): 232 (3.83), 299 (3.79) nm; EI-MS *m*/*z* (%): 332 ([M]^+^, 100), 317 ([M–CH_3_]^+^, 6), 273 (7), 249 (93), 233 (26), 207 (15), 203 (45), 161 (17); HR-EI-MS [M]^+^
*m*/*z* 332.1977 (calcd for C_20_H_28_O_4_ 332.1988).

5α,8-Epoxy-12-hydroxy-7,8-secoabieta-8,11,13-trien-7-al (**6**). Gum; [α]^25^_D_: +22.8 (*c* 0.9, CHCl_3_); IR ν_max_: 3423, 1706, 1434, 1255, 1169, 1096, 864, 738 cm^−1^; UV (MeOH) λ_max_ (log *ε*): 232 (3.75), 304 (3.79) nm; EI-MS *m*/*z* (%): 316 ([M]^+^, 100), 301 ([M–CH_3_]^+^, 4), 287 (7), 273 (13), 257 (9), 233 (73), 203 (53), 191 (23), 149 (11), 59 (11); HR-EI-MS [M]^+^
*m*/*z* 316.2033 (calcd for C_20_H_28_O_3_ 316.2039).

### 3.4. Antifungal Assay

The antifungal assay was performed in this study to evaluate the antifungal activity of diterpenoids isolated from the bark of *C. japonica* based on the methods used in our previous studies with slight modifications, including the fungus species and positive control [33]. *P. noxius*, an aggressive and destructive pathogen that can cause brown root disease in infected trees, was used in antifungal assays. Antifungal assessment of the isolated compounds was conducted using a mycelial radial growth inhibition technique against *P. noxius*. The tested compounds were added to sterilized potato dextrose agar (PDA) to give 100 ppm concentrations of extractives. The testing plates were incubated at 27 ± 2 °C. When the mycelium of fungi reached the edge of the control plate, the antifungal index was calculated as follows: antifungal index (%) = (1 − D_a_/D_b_) × 100, where D_a_: diameter of growth zone in the experimental dish (cm), D_b_: diameter of growth zone in the control dish (cm). The assays were performed three times, and the data were averaged. The commercial fungicide, didecyldimethylammonium chloride (DDAC), was used as a positive control at the concentration of 10 µg/mL.

## 4. Conclusions

In this study, five new abietane diterpenoids, 5-*epi*-12-hydroxy-6-*nor*-5,6-secoabieta-8,11,13-trien-7,5-olide (**1**), 12-hydroxy-6β-methoxy-6,7-secoabieta-8,11,13-trien-7,6-olide (**2**), 6β,12-dihydroxy-7,8-secoabieta-8,11,13-trien-7,8-olide (**4**), 5,12-dihydroxy-7,8-secoabieta-8,11,13-trien-7,8-olide (**5**), and 5α,8-epoxy-12-hydroxy-7,8-secoabieta-8,11,13-trien-7-al (**6**), together with one known abietane diterpenoid, obtuanhydride (**3**), were isolated and characterized from the bark of *C. japonica*. At a concentration of 100 µg/mL, the antifungal activities of compounds **5** and **6** against *P. noxius* were stronger than those of compounds **1**–**4**, with antifungal indices of 37.2% and 46.7%, respectively. The present findings revealed that compounds **5** and **6** have the potential to be used as natural antifungal agents against *P. noxius*.

## Figures and Tables

**Figure 1 plants-13-01197-f001:**
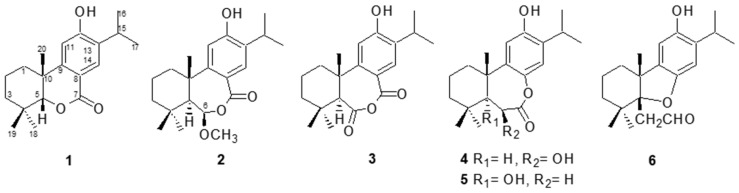
Structures of compounds **1**–**6**.

**Figure 2 plants-13-01197-f002:**
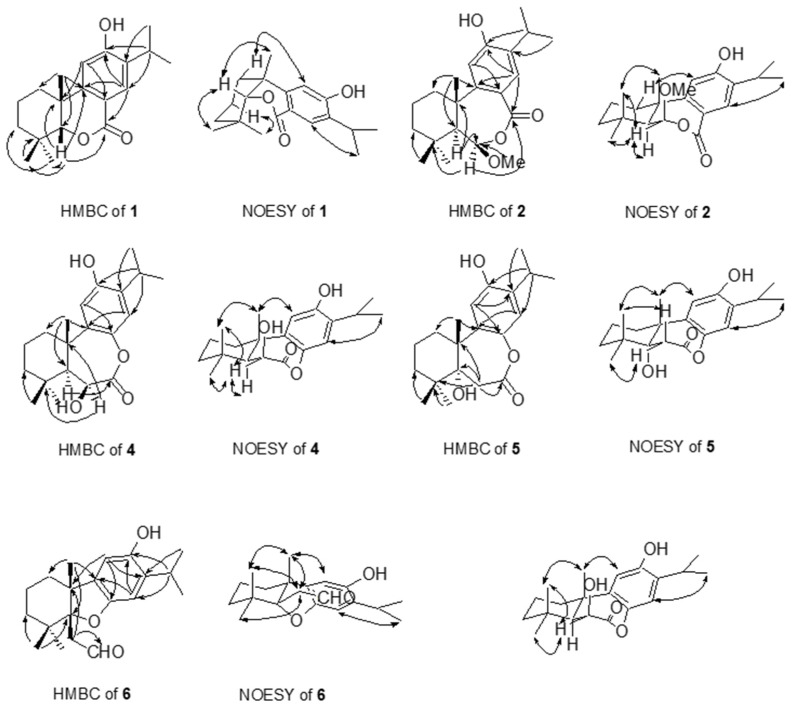
Selected HMBC and NOE correlations of **1**, **2**, and **4**–**6**.

**Table 1 plants-13-01197-t001:** ^1^H NMR spectral data of compounds **1**, **2**, and **4**–**6** (400 MHz in CDCl_3_).

No.	1	2	4	5	6
1	1.42 m, 2.36 m	1.82 m, 1.78 m	1.74 m, 1.77 m	1.39 m, 1.62 br d (13.2)	1.32 m, 1.72 td (12.8, 3.2)
2	1.40 m, 1.41 m	1.67 m, 1.80 m	1.62 m, 1.66 m	1.32 m, 1.40 m	1.16 m, 1.38 m
3	1.27 m, 1.43 m	1.23 m, 1.54 br d (12.8)	1.25 m, 1.53 br d (14.0)	1.34 m, 1.71 td (12.4, 3.6),	1.49 m, 1.57 m
5	4.05 s	1.69 d (1.6)	1.62 s		
6		5.11 d (1.6),	4.53 s	2.57 d (16.8), 2.61 d (16.8)	2.61 d (3.6)
7					9.38 t (3.6)
11	6.65 s	6.69 s	6.71 s	6.46 s	6.38 s
14	7.91 s	7.65 s	6.97 s	6.72 s	6.65 s
15	3.14 sept (7.2)	3.11 sept (6.8)	3.11 sept (6.8)	3.13 sept (6.8)	3.13 sept (6.8)
16	1.25 d (7.2)	1.25 d (6.8)	1.23 d (6.8)	1.21 d (6.8)	1.18 d (6.8)
17	1.26 d (7.2)	1.25 d (6.8)	1.22 d (6.8)	1.20 d (6.8)	1.17 d (6.8)
18	0.35 s	0.91 s	1.00 s	1.20 s	1.17 s
19	1.08 s	1.12 s	1.06 s	1.16 s	1.05 s
20	1.28 s	1.44 s	1.45 s	1.37 s	1.38 s
11-OH	5.71 s	5.24 s	4.94 br s		5.11 s
6-OCH_3_		3.34 s			

**Table 2 plants-13-01197-t002:** ^13^C NMR spectral data of compounds **1**, **2**, and **4**–**6** (100 MHz in CDCl_3_).

No.	1	2	4	5	6
1	35.2	40.0	39.6	40.8	38.1
2	18.7	19.0	18.9	17.9	18.1
3	40.5	41.5	41.6	38.4	41.0
4	37.2	35.0	35.7	37.4	37.0
5	91.6	61.4	67.6	96.0	96.3,
6		105.1	73.4	36.6	47.0
7	164.6	170.1	171.1	172.4	200.8
8	118.4	124.2	143.7	147.3	149.7
9	144.3	148.9	137.3	136.8	136.4
10	37.1	39.8	38.4	48.6	47.9
11	109.0	110.4	111.4	109.2	108.6
12	157.4	155.7	149.4	147.7	147.0
13	132.7	131.6	133.9	133.7	134.0
14	129.0	130.0	117.9	107.9	107.3
15	27.2	27.1	27.2	27.6	27.4
16	22.7	22.8	22.5	22.9	23.1
17	22.7	22.7	22.7	23.0	23.0
18	22.0	32.8	32.6	28.5	28.1
19	30.9	22.6	22.6	26.0	25.6
20	34.0	22.8	20.3	19.9	19.5
6-OCH_3_		56.5			

**Table 3 plants-13-01197-t003:** Antifungal index of diterpenoids from *C. japonica* bark against *Phellinus noxius.*

No.	1	2	3	4	5	6	DDAC
Antifungal index (%)	4.5	11.3	3.4	18.7	37.2	46.7	51.1

## Data Availability

Data are contained within the article and Appendix A.

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
