# Peer review of "Abietane Diterpenoids from the Bark of *Cryptomeria japonica* and Their Antifungal Activities against Wood Decay Fungi"

_plants, 2024, doi:10.3390/plants13091197_

Round 1
Reviewer 1 Report
Comments and Suggestions for Authors
The manuscript is original and reports phytochemical investigation of the bark of Cryptomeria japonica that led to the isolation of five new abietane diterpenoids. The structures were elucidated by analysis of spectroscopic data and comparison with spectral data of known analogues and their antifungal activities against wood decay fungi.
The author should give more explanation and citation about antifungal assay.
Change sesquarterpenoids per sesquirterpenoids, line 47 page 2.
Improve conclusion about antifungal assay.
Author Response
We’d like to appreciate your kind care on our manuscript (Manuscript ID: plants-2962110) and all the excellent comments and suggestions. We have modified the manuscript point-by-point according to your comments as listed below and highlighted the corrections with red color on the manuscript.
Response to Reviewer 1:
- The author should give more explanation and citation about antifungal assay.
Response: Thank you for your suggestions. We have revised them on page 8.
- Change sesquarterpenoids per sesquirterpenoids, line 47 page 2.
Response: We feel sorry and do not make any correction. Sesquiterpenoids are C15 terpenoids (reference 2). Sesquarterpenoids are C35 terpenoids (reference 29).
- Improve conclusion about antifungal assay.
Response: We have modified them on page 8.
Reviewer 2 Report
Comments and Suggestions for Authors
This paper deals with the identification of 5 new diterpenoids from the bark of the japanese cypress Cryptomeria japonica D. Don, more accurately abietanes. These diterpenoids were tested for the efficiency in preventing the growth of the tree pathogen fungus Phellinus noxius.
6 diterpenoids were isolated from bark methanol extracts, and submitted to HRMS, UV, IR and NMR. Their structures were established and 5 of them turned to be new compounds, whereas the last was obituanhydride. They had variable efficiencies against P. noxius, the most efficient exhibiting around 50% of growth inhibition at 100 µg/mL.
This paper was clear and did not split hairs. It presented with required details the determination of the structures of the isolated compounds, and a biological activity related to their hypothetical natural functions.
I have two major remarks:
* in the study about the antifungal activity, there is a lack of a positive control. It would be better to add a control with a known antifungal compound at 100 µg/mL (azoxystrobin for instance), in order to really assess the efficiency of the isolated compounds.
* the authors have been working for a long time on isolation of compounds from C. japonica bark, especially abietane diterpenoids (4 papers). It would be nice to highlight what is new in this study, for instance to answer to the following questions : were the new compounds in a fraction that had not been studied ? were they minor compounds or major ones ? were they found in some new samples that were enriched in them ?
I also have a minor remark: in Fig. 1, the structure of 2 and 3 should be drawn separately: it is strange and confusing to have a R moiety with a double bond supposedly for 2 and 3, that turns to be two single bonds (H and OCH3) for the compound 2.
Author Response
We’d like to appreciate your kind care on our manuscript (Manuscript ID: plants-2962110) and all the excellent comments and suggestions. We have modified the manuscript point-by-point according to your comments as listed below and highlighted the corrections with red color on the manuscript.
Response to Reviewer 2:
- * in the study about the antifungal activity, there is a lack of a positive control. It would be better to add a control with a known antifungal compound at 100 µg/mL (azoxystrobin for instance), in order to really assess the efficiency of the isolated compounds.
Response: Thank you for your suggestions. We have included the data of positive control, didecyldimethylammonium chloride (DDAC) on pages 6 and 8. Due to the antifungal index of DDAC is more than 100% at the concentration of 100 µg/mL, so it was tested at a concentration of 10 µg/mL and showed an antifungal index of 51.1%.
2.* the authors have been working for a long time on isolation of compounds from C. japonica bark, especially abietane diterpenoids (4 papers). It would be nice to highlight what is new in this study, for instance to answer to the following questions : were the new compounds in a fraction that had not been studied ? were they minor compounds or major ones ? were they found in some new samples that were enriched in them ?
Response: The reported new compounds were present in several subfractions that we have not yet investigated. Generally, they are minor compounds in this plant. These new compounds may be major compounds in other plant materials that have not yet been studied.
- I also have a minor remark: in Fig. 1, the structure of 2 and 3 should be drawn separately: it is strange and confusing to have a R moiety with a double bond supposedly for 2 and 3, that turns to be two single bonds (H and OCH3) for the compound 2.
Response: Thank you for your suggestions. We have modified them in Figure 1.
Reviewer 3 Report
Comments and Suggestions for Authors
The text below contains comments on manuscript entitled “Abietane diterpenoids from the bark of Cryptomeria japonica and their antifungal activities against wood decay fungi”.
The manuscript is focused on the investigation of the phytochemical profile of Cryptomeria japonica’s bark by isolation of new compounds by column chromatography, and further structure elucidation by HPTLC, 1- and 2-D NMR.
To my opinion the manuscript has a logical experimental design and the results fully explain the aim of the study.
I have a minor comment and would like to ask the authors to include the NMR settings, as well as, some peculiarities in the NMR spectra reading in material and methods, e.g. if there is phase-, base line corrections, scaling and the software for reading the spectra.
Comments on the Quality of English LanguageMinor editing of English language required
Author Response
We’d like to appreciate your kind care on our manuscript (Manuscript ID: plants-2962110) and all the excellent comments and suggestions. We have modified the manuscript point-by-point according to your comments as listed below and highlighted the corrections with red color on the manuscript.
Response to Reviewer 3:
I have a minor comment and would like to ask the authors to include the NMR settings, as well as, some peculiarities in the NMR spectra reading in material and methods, e.g. if there is phase-, base line corrections, scaling and the software for reading the spectra.
Response: We almost follow the standard operation procedure provided by the Varian NMR company to collect 1H and 13C NMR and 2D NMR spectra. For example, the 1H-NMR spectra were collected on a Varian-Unity-Plus-400 MHz spectrometer at a resonance frequency of 400.41 MHz equipped with a 5 mm Varian 400 ASW 1H/13C/31P/15N/4NUC PFG 40–162 MHz probe at 27 °C. The spectra were acquired with the parameters as follows: a 90 °C pulse (11.5 μs pulse width), relaxation delay of 5 s, acquisition time of 2.6 s, a spectral width of 6406.1 Hz (−3.3 to 12.7 ppm), with 16,384 complex data points, a 20 Hz spin. The scan number was between 20 and 256, which depended on the concentration of sample. The spectra were Fourier-transformed using MestReNova (version 14.1.2, Mestrelab Research, Escondido, CA, USA), including phasing and baseline correction. The area of the peaks was calculated by fitting Lorentzian–Gaussian peaks to the regions of investigation. Chemical shifts are referenced to residual solvent signals.